# Combination Therapy with a Bispecific Antibody Targeting the hERG1/β1 Integrin Complex and Gemcitabine in Pancreatic Ductal Adenocarcinoma

**DOI:** 10.3390/cancers15072013

**Published:** 2023-03-28

**Authors:** Tiziano Lottini, Claudia Duranti, Jessica Iorio, Michele Martinelli, Rossella Colasurdo, Franco Nicolás D’Alessandro, Matteo Buonamici, Stefano Coppola, Valentina Devescovi, Vincenzo La Vaccara, Alessandro Coppola, Roberto Coppola, Elena Lastraioli, Annarosa Arcangeli

**Affiliations:** 1Department of Experimental and Clinical Medicine, Section of Internal Medicine, University of Florence, 50134 Firenze, Italy; 2Department of Medical Biotechnologies, University of Siena, 53100 Siena, Italy; 3Physics of Life Processes, Huygens-Kamerlingh Onnes Laboratory, Leiden University, Niels Bohrweg 2, 2333 CA Leiden, The Netherlands; 4General Surgery Unit, Department of Medicine, Fondazione Policlinico Universitario Campus Bio-Medico, Via Alvaro del Portillo, 00128 Rome, Italy; 5Department of Surgery, University Roma La Sapienza, 00185 Rome, Italy

**Keywords:** PDAC, K^+^ channels, engineered antibodies, xenograft, ultrasound, photoacoustic imaging

## Abstract

**Simple Summary:**

Pancreatic ductal adenocarcinoma (PDAC) is one of the deadliest cancers and is forecasted to become the second most common cause of cancer-related deaths by 2030. Its therapy has proven extremely difficult and, consequently, there is an urgent need for novel therapeutic strategies for PDAC. Although Gemcitabine chemotherapy has long been used as a standard of care for PDAC, it suffers from limited efficacy and high toxicity. Here, we describe a new therapeutic strategy based on a single chain bispecific antibody (scDb-hERG1-β1) which targets a cancer-specific antigen, i.e., the complex formed by the K^+^ channel hERG1 and the β1 integrin (hERG1/β1 integrin complex). The combination of scDb-hERG1-β1 with sub-optimal doses of Gemcitabine in mice implanted with PDAC showed good therapeutic efficacy, low toxicity and, consequently, prolonged survival time. Our data pave the way for improving the therapy of PDAC, and possibly other cancers, by combining chemotherapy with ion channel modulators.

**Abstract:**

Pancreatic ductal adenocarcinoma (PDAC) represents an unmet medical need. Difficult/late diagnosis as well as the poor efficacy and high toxicity of chemotherapeutic drugs result in dismal prognosis. With the aim of improving the treatment outcome of PDAC, we tested the effect of combining Gemcitabine with a novel single chain bispecific antibody (scDb) targeting the cancer-specific hERG1/β1 integrin complex. First, using the scDb (scDb-hERG1-β1) in immunohistochemistry (IHC), Western blot (WB) analysis and immunofluorescence (IF), we confirmed the presence of the hERG1/β1 integrin complex in primary PDAC samples and PDAC cell lines. Combining Gemcitabine with scDb-hERG1-β1 improved its cytotoxicity on all PDAC cells tested in vitro. We also tested the combination treatment in vivo, using an orthotopic xenograft mouse model involving ultrasound-guided injection of PDAC cells. We first demonstrated good penetration of the scDb-hERG1-β1 conjugated with indocyanine green (ICG) into tumour masses by photoacoustic (PA) imaging. Next, we tested the effects of the combination at either therapeutic or sub-optimal doses of Gemcitabine (25 or 5 mg/kg, respectively). The combination of scDb-hERG1-β1 and sub-optimal doses of Gemcitabine reduced the tumour masses to the same extent as the therapeutic doses of Gemcitabine administrated alone; yielded increased survival; and was accompanied by minimised side effects (toxicity). These data pave the way for a novel therapeutic approach to PDAC, based on the combination of low doses of a chemotherapeutic drug (to minimize adverse side effects and the onset of resistance) and the novel scDb-hERG1-β1 targeting the hERG1/β1 integrin complex as neoantigen.

## 1. Introduction

Pancreatic ductal adeno carcinoma (PDAC) is one of the deadliest cancer types, accounting for 4.7% of all deaths from cancer globally [1,2]. Its incidence has been gradually rising in recent years, unlike many other cancers with declining incidence and mortality. The burden of the disease includes 495,748 new cases and 466,003 related deaths per annum. Hence, incidence and mortality overlap substantially for PDAC. Indeed, the median life expectancy is <1 year for patients with metastatic PDAC. Therefore, there is an urgent need for novel therapeutic strategies for PDAC. It is noteworthy that the dismal prognosis of PDAC can be traced back to its late detection, intrinsic chemoresistance and treatment failure. At present, PDAC therapy includes surgery for resectable tumours and chemotherapy (with Gemcitabine, NP-paclitaxel or FOLFIRINOX) for patients with advanced PDAC [3].

Since its approval in 1995, Gemcitabine has been the most common chemotherapeutic treatment for PDAC, although its delivery into the tumour mass is often impaired by the desmoplastic stroma, which characterizes the tumour microenvironment in PDAC [4]. Moreover, several mechanisms, including drug efflux pumps and transporters and other detoxifying mechanisms can impair the efficacy of Gemcitabine to exert its antineoplastic effects on DNA [5]. To compensate, Gemcitabine is administered in high and repeated doses, which can generate serious side effects including bone marrow suppression, liver and kidney failure, rash and hair loss. On the other hand, the use of low doses of the chemotherapeutic drug can provide a reduction in the toxic side effects but at the expense of the antineoplastic efficacy. Currently, to overcome the lack of specificity of chemotherapeutic drugs and their related toxic side effects, novel treatments and mechanisms are being widely studied [3]. Targeted therapies use pharmacological agents (i.e., monoclonal antibodies) that inhibit growth, increase cell death and interfere with metastasis [4]. Targeting specific cancer related proteins may be crucial for increasing the treatment efficacy as well as reducing the toxicity in normal cells. In this regard, combinatory treatments are becoming increasingly common [5]. Overall, the identification of novel therapeutic targets and novel treatment regimens, including combinations, are strongly encouraged in PDAC.

For some decades, mounting evidence has pointed to ion channels as novel biomarkers in human cancers [6]. Among them, potassium channels exert a key role [7]. In particular, the human *ether-á-go-go–related gene* (hERG1) is expressed in different types of human solid cancers, while absent in normal tissues [8,9,10,11,12,13,14,15,16]. In fact, hERG1 is physiologically expressed in excitable tissues, particularly in the heart where it regulates the repolarization phase of the cardiac action potential [17]. Moreover, hERG1 is functionally expressed in different neuronal populations where it modulates electrical excitability [18], muscle cells where it plays a role in regulating contractility [19], as well as in endocrine cells where it acts as a regulator of hormone secretion through modulation of action potential frequency [20,21]. In tumours, the presence of hERG1 channels contributes (i) to maintain the resting potential at relatively less negative values compared to normal cells which seem to be essential for unlimited growth [22], and (ii) to trigger intracellular signalling pathways involved in cell survival, proliferation, motility and invasion [23]. This occurs through the formation of a molecular complex with the β1 subunit of the integrin receptors [24]. Overall, hERG1 could represent a novel cancer biomarker and a therapeutic target in different tumours [16,25,26,27,28,29]. However, the strong physiological expression of hERG1 in the human cardiomyocytes limits if not eliminates the possibility of targeting the channel as it is [30]. To overcome this hindrance, we searched for either functional and/or molecular differences between the “cardiac” and the “tumour” hERG1. Indeed, it emerged that, in tumours, hERG1 resides in a novel conformational state, strictly bound to the β1 subunit of the integrin adhesion receptors (β1 integrin), within a macromolecular complex where the two proteins are at a distance of less than 1 nm. This does not occur in the heart where hERG1 is bound to classical accessory subunits, such as the “potassium voltage-gated channel subfamily E regulatory subunit 1” (KCNE1) [23]. These findings raised the strong possibility for the hERG1/β1 integrin complex to be considered as a druggable novel oncological target.

Accordingly, we generated (and patented—Section 6) a novel, specifically engineered, single chain, bispecific antibody in the form of a diabody (scDb-hERG1-β1) which has proven effective in targeting the hERG1/β1 integrin complex in cancer cells with high affinity [31]. Blocking this complex switches the PI3K/Akt pathway off and this has a negative impact on cell growth, angiogenesis and metastatic progression. The diabody is also characterized by desirable pharmacodynamic parameters, such as rapid clearance [32], which contribute to making the scDb-hERG1-β1 a potential candidate for targeted therapy.

In the present study, we evaluated the efficacy of a combination therapy employing the scDb-hERG1-β1 with Gemcitabine as the basis of a possible novel therapeutic strategy for PDAC.

## 2. Materials and Methods

### 2.1. In Silico Analysis

An in silico analysis was carried out using UCSC Xena Browser (University of California, Santa Cruz, http://xena.ucsc.edu/ (accessed on 25 January 2023)) [33]. This browser allows the direct comparison of the expression of genes of interest in tumour datasets stored in the TCGA library with healthy samples from the GTEx database (https://gtexportal.org/home/ (accessed on 25 January 2023)) [34]. Specifically, we filtered the TCGA data in order to keep only samples derived from PDAC while, for the control group, healthy pancreas samples retrieved from GTEx were used. This led to a final comparison between 179 tumour samples and 167 normal tissues.

### 2.2. Sample Collection and Clinicopathological Characteristics

Patients were treated at Campus Bio-Medico University (Rome, Italy). Tissues were collected after informed written consent was obtained, and approval of the local ethics committee (PAR: 13.21) granted. PDAC samples and the corresponding normal tissues were then analysed.

### 2.3. Immunohistochemistry (IHC)

Forty formalin-fixed, paraffin-embedded PDAC samples were analysed for the expression of the following proteins: hERG1, β1 integrin and KCNE1. A total of 132 formalin-fixed, paraffin-embedded PDAC samples were analysed for the expression of the hERG1-β1 integrin complex (commercial tissue microarray number: PA2082a, BioMax), and 44 PDAC cases derived from the series of an earlier study [14] were analysed for the expression of hERG1-β1 integrin complex. IHC was carried out on 7 μm sections on positively charged slides. After dewaxing and rehydrating the sections, endogenous peroxidases were blocked with a 1% H_2_O_2_ solution in phosphate-buffered saline (PBS). Subsequently, antigen retrieval was performed with different procedures, depending on the antibody used: (1) by treatment with proteinase K (5 μg/mL) in PBS at 37 °C for 5 min for hERG1 and scDb-hERG1/β1 (MCK Therapeutics Srl, Florence, Italy); (2) by heating the samples in a microwave oven at 600 W in citrate buffer pH 6.0 for 20 min (for β1 integrin and KCNE1 staining). The following antibodies were used: anti-hERG1 monoclonal antibody (MCK Therapeutics Srl; 0.005 μg/μL), anti-BETA1 integrin (monoclonal antibody (4B7R) to Integrin β1, Abcam, Cambridge, UK, 1:35), anti-KCNE1 (Abcam, Cambridge, UK). Incubation with the primary antibodies was carried out overnight at 4 °C, except for anti–β1 integrin antibody, which was incubated for 2 h at room temperature. Immunostaining was performed with a commercially available kit (PicTure max kit; Invitrogen, Carlsbad, CA, USA) according to the manufacturer’s instructions.

### 2.4. Western Blot (WB) and Co-Immunoprecipitation (Co-IP)

All procedures were carried out at 4 °C. Samples were homogenized in cold protein extraction buffer (1 × cell lysis buffer) and sonicated for 30 min. For immunoprecipitation, total lysates (0.5 mg) were subjected to a preclearing step by incubating them with Protein A/G Plus-Agarose for 2 h at 4 °C. Protein extraction, quantification and total lysate incubation with protein A/G agarose beads (Santa Cruz Biotechnology, Dallas, TX, USA) were performed as previously reported [30]. LEAF Purified anti-human, Clone TS2/16 (BioLegend, San Diego, CA, USA) was used to immunoprecipitate the β1 integrin. After overnight incubation, the immuno-complex was captured by adding 30 μL of protein A/G agarose beads for 2 h at 4 °C (with rolling agitation). The agarose beads were washed 3 times in ice-cold wash buffer and 3 times in ice cold PBS followed by addition of 2 × Laemmli buffer (10 μL) and boiled for 5 min at 95 °C. Afterwards, SDS-PAGE was performed with total lysates and immunoprecipitates (IPs). After electrophoresis, proteins were transferred onto PVDF membrane (previously activated) in blotting buffer under cold condition for 1 h at 100 V. The PVDF membrane was then blocked with 5% BSA in T-PBS (0.1% tween) solution for 3 h at room temperature to cover the unspecific antibody binding sites on the membrane. Next, incubations with antibodies and co-IPs on primary human samples were performed as previously described [30]. Western blotting was performed on co-IPs and total lysates with the following antibodies: polyclonal rabbit polyclonal antibody against hERG1 C terminus (hERG1 CT pan–polyclonal antibody; DT-552, Di.V.A.L. Toscana Srl, Sesto Fiorentino, Italy), 1:1000; rabbit polyclonal antibody against β1 integrin C terminus (RM12, Immunological Sciences), 1:1000; mouse monoclonal antibody against KCNE1 (Abcam), 1:500; and mouse monoclonal antibody against tubulin (Santa Cruz Biotechnology, Dallas, TX, USA).

### 2.5. Cell Culture

Cells were cultured at 37 °C with 5% CO_2_ in a humidified atmosphere. PANC-1 were cultured in DMEM (Euroclone, Milan, Italy) supplemented with 4 mM of L-glutamine and 10% FBS (Euroclone). MiaPaca-2 and BxPc-3 were cultured in RPMI (Euroclone) supplemented with 2 mM of L-glutamine and 10% FBS (Euroclone). PSC-RLT were cultured in DMEM F-12 (Euroclone) supplemented with 2 mM of L-glutamine and 10% FBS (Euroclone). HPDE were cultured in 50% RPMI 1640 (Life Technologies, Carlsbad CA, USA), 50% Keratinocyte medium—SFM (Life Technologies) supplemented with FBS 10% heat inactivated, MEM Non-Essential Amino Acids 1 × (Life Technologies), Pen/Strep 1X, Hepes 10 mM (Life Technologies), bovine pituitary extract 0.025% (Life Technologies), and EGF human recombinant 2.5 ug/L (Life Technologies).

PANC-1, MiaPaca-2 and BxPc-3 cells were obtained from the American Type Culture Collection (ATCC); HPDE were kindly gifted by Prof. I. Szabò (University of Padua, Italy); PSC-RLT cells were kindly gifted by Prof. F. Alves (UMG, Department of Hematology and Medical Oncology and the Institute for Diagnostic and Interventional Radiology, Goettingen, Germany). When cultured as spheroids, cells were seeded on an agarose base layer (1.5 g/L) in 96 wells plates and grown for 72 h.

### 2.6. Cell Viability Assay

Cell viability was measured by the trypan blue (Sigma, Darmstad, Germany) exclusion test as in [35]. After incubation with the drug and the scDb-hERG1-β1 antibody, the trypan blue dye was added to the cells and live cells were counted using LUNA-II™ Automated Cell Counter (Logos Biosystems, Villeneuve d’Ascq France). The 50% inhibitory concentration (IC_50_) was calculated using the equation $Y=\mathrm{Min}+\frac{Max-min}{1-{\left(\frac{X}{IC50\;}\right)}^{Hill}coefficient}\;$ as in [36].

The combination index was determined using Compusyn software 1.0 [36].

### 2.7. scDb-hERG1-β1 Antibody Production and Purification

The scDb-hERG1-β1 antibody was expressed in yeast cells as described previously [31,37], by performing an induction with methanol and harvesting the supernatant in which the protein is secreted for subsequent purification. Purification of scDb-hERG1-β1 was performed by affinity chromatography, using an ÄKTA protein purification system (Cytiva, Marlborough, MA, USA) with a HisTrap HP 5 mL column as previously described [31,37]. Elution fractions in which the scDb-hERG1-β1 protein was detected were then collected, pooled and dialyzed into a PBS using a Slide-A-Lyzer™ dialysis cassette (Thermo Fisher, Waltham, MA, USA). The protein concentration was quantified by NanoDrop (Thermo Scientific, Waltham, MA, USA).

### 2.8. Labelling of the scDb-hERG1/β1 with Alexa 488 and Indiocyanine Green (ICG)

A mass of 150 μg of scDb-hERG1/β1 (MCK Therapeutics srl, www.mcktherapeutics.com (accessed on 25 January 2023); info@mcktherapeutics.com) at a concentration of 2 mg/mL in PBS solution and 0.1 M sodium bicarbonate buffer (pH 8.3) was incubated for 1 h at 22 °C in agitation with 12 μL of Alexa Fluor 488 (Succinimidyl Ester; Thermo Fisher Scientific, Waltham, MA, USA) and resuspended in DMSO at 10 mg/mL. The reaction was blocked for 5 min in ice and the labelled protein was purified by size exclusion chromatography on a Sephadex G25 (Sigma) column equilibrated with PBS. The amine-reactive dye ICG was first dissolved in anhydrous DMSO (Sigma-Aldrich, Saint Louis, MO, USA). Solutions of scDb-hERG1/β1 were incubated with ICG at molar ratios of ICG:scDb of 5, 10, and 20 (5×, 10×, and 20×) in conjugation buffer (0.002 M NaHCO_3_ + 0.048 M Na_2_CO_3_ + 0.15 M NaCl, pH 8.5) for 1 h in the dark at 37 °C with gentle mixing at 750 rpm. Total volume (250 μL) and DMSO percentage were constant for all reactions. Purification was performed using PD-10 Desalting Columns (Cytiva, Marlborough, MA, USA) according to the manufacturer’s instructions.

### 2.9. Immunofluorescence (IF)

IF on cells was performed following the protocol previously described [31]. For IF with bispecific antibody, after 2 h of blocking in PBS with 10% BSA, sections were incubated overnight with scDb-hERG1-β1-Alexa488 (20 µg/mL final concentrations). All incubations were performed at 4 °C. The following day, slides were incubated with Hoechst (1:1000 in PBS, 45 min; Merck Sigma, Burlington, MA, USA), to stain cell nuclei. Images were captured using confocal microscope, Nikon TE2000 (Tokyo, Japan).

### 2.10. Mouse In Vivo Model

The orthotopic xenograft model was obtained by the injection of PANC-1 cells into the mice pancreas. The ultrasound (US)-guided injection method detailed in [38] was followed using the VevoLAZR-X imaging system (Fujifilm Visualsonics, Toronto, ON, Canada). The injection was performed into female athymic Foxn1nu/nu mice (6 weeks; Envigo, Indianapolis, IN, USA). PANC-1 cells were cultured in DMEM + 10% FBS medium, under the conditions of 37 °C and 5% CO_2_. For the injection, 1 × 10^6^ tumour cell lines were suspended in 20 µL of PBS. To evaluate the therapeutic effect of the combination of the scDb-hERG1-β1 with Gemcitabine, mice were divided into 5 groups of treatment, 8 days after the cell injection: (i) control (saline; *n* = 16); (ii) scDb-hERG1-β1 (16 mg/kg; *n* = 8); (iii) Gemcitabine (5 mg/kg; *n* = 11); (iv) Gemcitabine (25 mg/kg; *n* = 7); and (v) scDb-hERG1-β1/16 mg/kg + Gemcitabine/5 mg/kg (*n* = 5). The treatments were administrated starting from day 8 and ended on day 36.

All the experiments were performed at L.I.Ge.M.A. laboratory (Laboratory of genetic engineering for the production of mouse models) at the Animal house (Ce.S.A.L) of the University of Florence. Mice were housed inside the sterile room in ventilated cabinets with a canonical 12-h dark-light cycle and unlimited access to food and water. The procedures received the approval from the Italian Ministry of Health with authorization n. 843/2020-PR. To assess the impact of treatments on survival, animals were monitored and euthanized when they showed signs of suffering. We performed Kaplan–Meier survival analysis to show the fraction of mice living for a certain amount of time after treatment.

### 2.11. Ultrasound (US) and Photoacoustic (PA) Imaging

Ultrasound imaging was performed with the VevoLAZR-X system [39]. A 55-Mhz transducer was used for the US-guided injection of PANC-1 cells into the tail of the pancreas, as well for the 3D-axial scan of tumour masses. The 3D acquisition was performed in B-mode using the 3D motor that allows the transducer to scan the tumour masses in various sections along the longitudinal axis. In each section, the tumour perimeter was bordered (region of interest—ROI) and then, by using VevoLAB software, the 3D rendering of the entire tumour masses was obtained. During the US-guided injection and the 3D-axial scan of tumours, mice were maintained anesthetized by 2% isoflurane on a pad heated at 37 °C. The scDb-hERG1-β1 conjugated with ICG (scDb-hERG1-β1-ICG) was administrated intravenously (iv) one hour before the imaging session. The PA signal of ICG in the pancreas was monitored in real time by PAI, performed with VevoLAZR-X. PA images were acquired with the 55-MHz linear array transducer.

### 2.12. Statistics

GraphPad Prism software (version 9.1, GraphPad Software, San Diego, CA, USA) was performed for statistical analysis and graphics. Statistically significant differences were determined by using one-way ANOVA plus unpaired Welch *T*-test for single treatment group comparison. A *p*-value of <0.05 was considered statistically significant. The log-rank test was performed to assess significance between mean survival in Kaplan–Meier curves, computed by R software (v.4.2.2). We adopted the mean survival time which represents a better predictor for general outcome in this setting [40].

## 3. Results

### 3.1. Expression of the hERG1/β1 Integrin Complex in PDAC

#### 3.1.1. Primary Human PDAC Samples

With the aim of targeting the hERG1/β1 integrin complex in PDAC, we began by determining the occurrence of the complex in primary PDAC samples and in PDAC cell lines. First, a bioinformatic analysis on different datasets referring to RNASeq data of PDAC, and its normal counterpart, was performed. We focused on the analysis of the two components of the hERG1/β1 integrin complex (i.e., hERG1 and the β1 subunit of integrin receptors). Hence, we analysed the expression levels of the genes encoding hERG1 (KCNH2), and the β1 integrin (ITGB1). The expression of the classical accessory subunit KCNE1, which is physiologically complexed with hERG1 in the heart [41] and is substituted by the β1 integrin in tumours [30], was also evaluated, by analysing the KCNE1 transcripts. It emerged that KCNH2 was more expressed in tumour than in normal tissues, whereas ITGB1 was expressed at the same levels in normal and tumour samples, while KCNE1 was expressed at lower levels, with no significant difference between healthy and tumour tissue (Figure 1A). We then determined the expression of the corresponding proteins—hERG1, β1 integrin and KCNE1—by IHC analysis of 40 samples of normal pancreatic tissues and PDAC. hERG1 was not detected in normal primary exocrine pancreas, whereas there was strong positivity in 72.5% (29/40) of PDAC samples. Positive expression of β1 integrin was observed in 42.5% (17/40) of normal pancreatic tissues and in 82.5% (33/40) of PDAC samples. KCNE1 was detected in 47.5% (19/40) of normal pancreatic tissue samples and in 12.5% (5/40) of PDAC samples. KCNE1 staining showed a more defined staining in normal mucosa, while in PDAC samples, the staining pattern was less evident. A statistically significant positive correlation emerged between hERG1 and β1 integrin in PDAC samples (R: 0.4835, *p* = 0.001; Pearson coefficient correlation) and between β1 integrin and KCNE1 in normal pancreatic tissue samples (R: 1.000, *p* < 0.0001; Pearson coefficient correlation) (Figure 1B and Table 1). Notably, hERG1 and β1 integrin co-localised in the same tumour cells in the PDAC tissue samples (see insets in Figure 1B).

These results are suggestive of the interaction of hERG1 with the β1 integrin in primary cancers, to form a hERG1/β1 integrin complex. This was confirmed by the co-immunoprecipitation of the two proteins in the same primary PDAC samples. In particular, we extracted proteins from primary PDAC and from the corresponding normal tissues taken at distance from the cancer. hERG1 was expressed at high levels in PDAC, while almost absent in normal tissue. KCNE1 was not expressed in cancer tissues but showed a significant expression in normal tissues (Figure 1C). Overall, WB data fully agreed with the IHC data. Furthermore, hERG1 and β1 co-immunoprecipitated only in PDAC samples (Figure 1D).

Finally, the occurrence of a hERG1/β1 integrin complex in primary PDAC was confirmed by IHC using a bispecific antibody, the scDb-hERG1-β1, which selectively recognizes the hERG1/β1 integrin complex [31,37]. In total, 132 primary PDAC samples were analysed and a high (>50% of the cells) scDb-hERG1-β1 staining was detected in 68.2% (90/132) of cases, with a statistically significant association with hERG1 staining (*p* < 0.0001) (Figure 2). Figure 2 shows the staining for hERG1-β1 (panel A) and hERG1 (panel B). The arrows show the same cells stained with scDb-hERG1-β1 and mAb-hERG1.

Overall, data obtained from primary human PDAC clearly indicate the presence of a hERG1/β1 integrin macromolecular complex in a high percentage of the samples, with a high score of labelling.

#### 3.1.2. PDAC Cell Lines

We then determined the presence of the hERG1/β1 integrin complex in different human PDAC cell lines (PANC-1, MiaPaca-2 and BxPC3) by IF using the scDb-hERG1-β1. The human pancreatic duct epithelial (HPDE) cells [42] and pancreatic stellate cells (PSC-RLT) cells [43] were used as controls. It emerged that PANC-1 and MiaPaca-2 were strongly positive for the hERG1-β1 complex showing clear membrane staining (Figure 3A). Such observations were confirmed by graphic quantification. PANC-1 and MiaPaca-2 showed a high signal compared to BxPC3, which although positive, showed low signal. In particular, both HPDE and PSC-RLT appeared negative. A clear scDb-hERG1-β1 fluorescent signal was also observed in PANC-1 when cultured in 3D as spheroids (Figure 3B).

### 3.2. Combination of scDb hERG1/β1 Integrin with Gemcitabine: In Vitro Data

The effects of Gemcitabine and of scDb-hERG1-β1 on the vitality of the cell lines described above were then determined. Appendix A shows the dose-response curves for Gemcitabine and scDb-hERG1-β1 in PANC-1, MiaPaCa-2, BxPC3 and PSC-RLT cells. These data allowed us to determine the IC_50_ values (Table 2). The efficacy of Gemcitabine was high (IC_50_ < 5 μM) for BxPC3, good (IC_50_ ~ 40 μM) for PANC-1 and MiaPaca-2 cells, and low for PSC-RLT (IC_50_ > 150 μM). The scDb-hERG1-β1 showed high efficacy (IC_50_ ~ 20 μg/mL) for PANC-1 and MiaPaca-2 cells, low efficacy (IC_50_ = 160 μg/mL) for BxPC3 and insignificant efficacy (IC_50_ > 500 μg/mL) for PSC-RLT. Overall, the scDb-hERG1-β1 IC_50_ values agree with the expression levels of the hERG1/β1 integrin complex in the pancreatic cell lines studied. Starting from the IC_50_ values experimentally determined, we used different doses (IC_25_, IC_75_ and IC_100_) of either Gemcitabine, scDb-hERG1-β1 or their combinations to measure the combination index (CI) as in [36] (Figure 4A). The CI values clearly indicate a synergistic effect of the scDb-hERG1-β1 and Gemcitabine combination but only in the PDAC cells which express the hERG1/β1 integrin complex (Table 2). The effects of the combination of the IC_50_ and IC_25_ doses of Gemcitabine with the IC_50_ of scDb-hERG1-β1 on PANC-1 cells are shown in Figure 4B.

### 3.3. Combination of the scDb hERG1/β1 Integrin with Gemcitabine: In Vivo Data

#### 3.3.1. Penetration of the scDb hERG1/β1 into the Tumour Masses of an Orthotopic Xenograft PDAC Mouse Model: Evidence with Photo Acoustic (PA) Imaging

Before testing the effects in vivo of the combination of the scDb-hERG1-β1 with Gemcitabine, we analysed the capacity of the antibody to reach the tumour masses, using the same animal model as for the pharmacological tests. To this purpose, we used the eco-guided xenograft model described in Lottini et al., 2021 [38], using PANC-1 cells. Cell engraftment and the development of the tumour masses were monitored by B-mode ultrasound 3D imaging, using the VevoLAZR-X imaging station (Figure 5A). After 8 days, tumour masses reached a volume of about 10 mm^3^ (the minimum for accurate visualization in the pancreas with ultrasound). Two mice with a tumour mass of 18–20 mm^3^ were injected intravenously with a bolus of 50 µL of scDb-hERG1-β1 conjugated with ICG and 50 µL of ICG alone, to be visualized by PA imaging. One hour after injection, we started the imaging session. A PA signal (green spots indicated by the white arrow in Figure 5B) was detected in the tumour mass of the mouse treated with scDb-hERG1-β1-ICG (Figure 5B). No signal within the tumour mass was observed in mice injected with ICG alone (Figure 5C). This indicates that the scDb-hERG1-β1 specifically penetrates the tumours.

#### 3.3.2. Effects of Gemcitabine in the Orthotopic Xenograft PDAC Mouse Model

In a first set of experiments, we tested the effects of Gemcitabine at the dose of 25 mg/kg and sub-optimal dose of 5 mg/kg. Eight days after cell injection, mice were randomized into three groups of treatment: (i) control (saline); (ii) Gemcitabine 5 mg/kg; and (iii) Gemcitabine 25 mg/kg. Both saline and gemcitabine were injected intraperitoneally (i.p.) three times per week. Tumour growth was monitored by US imaging, for 4 weeks from the beginning of the treatments (Figure 6B). At the end point, the tumour size of the control group reached a mean value of 246.2 mm^3^ (Figure 6A). A similar trend of growth was observed in the group treated with Gemcitabine 5 mg/kg, which, however, reached a lower mean volume (189.2 mm^3^); the difference was not statistically significant. Gemcitabine at 25 mg/kg had greater impact on tumour growth, with a mean value at the end point of 89.4 mm^3^, which was significantly different from the control group (*p* = 0.0004, Figure 6A). Some representative US images of PDACs treated with saline and different doses of Gemcitabine are shown in Figure 6B.

#### 3.3.3. Effects of Gemcitabine in Combination with scDb-hERG1-β1

We next studied the therapeutic efficacy of scDb-hERG1-β1, alone or in combination with Gemcitabine, using the same experimental model described above. Eight days after US-guided cell injection in the pancreas, mice were randomized into five groups: control (saline), Gemcitabine 25 mg/kg, Gemcitabine 5 mg/kg, scDb-hERG1-β1, Gemcitabine 5 mg/kg + scDb-hERG1-β1. The scDb-hERG1-β1 was administered daily by intravenous injection at the dose of 16 mg/kg, as previously described [31]. The scDb-hERG1-β1 was combined only with sub-optimal doses of Gemcitabine. In the mice group treated with scDb-hERG1-β1, the trend of tumour growth was similar to the group treated with 5 mg/kg Gemcitabine with no statistically significant difference. Moreover, no statistically significant difference emerged with the tumour volumes of the control group at the end point (day 36) (control = 246.2 mm^3^; scDb-hERG1-β1 = 206.3 mm^3^; *p* = 0.5) (Figure 7A). Conversely, the combination of 5 mg/kg Gemcitabine and scDb-hERG1-β1 strongly potentiated the effect of the chemotherapeutic drug. Indeed, a statistically significant reduction in tumour volume emerged at day 36 (control = 246.2 mm^3^, scDb-hERG1-β1 + 5 mg/kg Gemcitabine = 93 mm^3^; *p* = 0.003) (Figure 7A). Representative images are in Figure 7B.

A Kaplan–Meier survival curve was then generated to evaluate the impact of the different treatments on survival (Figure 8).

The control and the 5 mg/kg Gemcitabine-treated groups showed a similar survival trend, thus reflecting the inefficacy of the treatment provided by the sub-optimal dose of Gemcitabine (Figure 8). Mice treated with the therapeutic dose (25 mg/kg) of Gemcitabine (blue line) started to die later, but later showed a narrower time window (days 62–88) of death, if compared to all the other groups (Figure 8). This fact is consistent with the high toxicity of Gemcitabine. Indeed, mice treated with 25 mg/kg of Gemcitabine showed severe signs of suffering, such as reduced motility and back arching, which induced us to apply euthanasia. In addition, this group and the control group showed decreased nest-building behaviour earlier than the other groups. In this group, the mean survival time was 77 days, longer than either the control group (67 days) or the group treated with Gemcitabine at a sub-optimal dose (59 days). The difference in mean survival time between the group treated with therapeutic doses (25 mg/kg) and sub-optimal doses (5 mg/kg) of Gemcitabine was statistically significant (*p* = 0.02)—Appendix A. An increased mean survival time was observed in the groups treated with scDb-hERG1-β1 (79 days) and in the group treated with scDb-hERG1-β1 in combination with 5 mg/kg Gemcitabine (82 days) compared to all the other groups. The mean survival time of the group treated with scDb-hERG1-β1 was significantly different from the mean survival time of the group treated with 5 mg/kg Gemcitabine (*p* = 0.05, Appendix A). The mean survival time of the group treated with the combination of sub-optimal doses of Gemcitabine in combination with scDb-hERG1-β1 was much higher than the mean survival time of the group treated with sub-optimal doses of Gemcitabine (82 days vs. 59 days, *p* = 0.1, Appendix A).

At the end point, mice treated with Gemcitabine at the therapeutic dose showed ascites and abnormal liver in 30% of cases (two out of seven) (Appendix A), confirming the side-effects of the treatment. In the groups treated with scDb-hERG1-β1 alone and scDb-hERG1-β1+ 5 mg/kg Gemcitabine, no ascites or abnormal livers were observed. Furthermore, the general signs of suffering, such as abnormal posture and reduced mobility, were observed later than in the control and the 25 mg/kg Gemcitabine groups.

## 4. Discussion

In the present paper, we evaluated the potential therapeutic effects of combining Gemcitabine with a novel single chain bispecific antibody (scDb-hERG1-β1) which targets the cancer-specific hERG1/β1 integrin complex. After proving the selective presence of the hERG1/β1 complex in primary PDAC samples and cell lines, and its absence in normal tissues and cells, we first tested the combination in vitro and in vivo. Gemcitabine and scDb-hERG1-β1 showed a good synergic activity on cell vitality of PDAC cells in vitro. The combination treatment was then tested in vivo, using an orthotopic xenograft mouse model, obtained by US-guided injection of PDAC cells. After verifying that scDb-hERG1-β1 readily penetrated tumour masses, we combined the antibody with sub-optimal (5 mg/kg) doses of Gemcitabine. This combination reduced the volume of the tumour masses to the same extent as a therapeutic dose of the drug and produced an increase in survival without significant toxic side effects.

The presence of the hERG1-β1 integrin complex in PDAC primary samples had previously been shown by co-IP in two earlier studies [24]—both the presence of the complex in PDAC cell lines [31] and the high-level expression of hERG1 in primary human PDAC samples [14]. Here, we confirmed and extended the evidence by (i) analysing a large cohort of cases, (ii) applying independent techniques (in silico, IHC, WB and co-IP), and (iii) ruling out the presence of the complex in normal pancreatic tissues. Furthermore, through IHC analysis, we provide evidence that a strong correlation exists between the expression of the hERG1 protein and the expression levels of the hERG1/β1 integrin complex. Applying the same antibodies to a smaller cohort of tissue samples, we also recently showed a good correlation between the co-expression of the two proteins in pancreatic neuroendocrine tumours [35]. The latter result is significant since it strengthens the case for the hERG1-β1 complex being a cancer-specific phenomenon, paving the way for clinical applications. Thus, the hERG1 mAb may be used in IHC for diagnostic and prognostic purposes. It can also serve as a predictive biomarker in clinical trials where scDb-hERG1-β1 (or other strategies targeting the hERG1/β1 integrin complex) may be applied therapeutically.

Moreover, we also provided evidence that the engineered scDb-hERG1-β1 can be used for in vivo imaging. Indeed, scDb-hERG1-β1 turned out to penetrate well into the tumour masses in mice after intravenous injection. Some examples of other antibodies conjugated and used for molecular imaging include ProstaScint which is a mAb used for diagnostic imaging of prostate cancer and detection of nodal metastases during “pre-prostatectomy” or recurrence in post-prostatectomy patients with a rising level of prostate-specific antigen. In addition, the therapeutic mAbs, cetuximab and trastuzumab, are used as in vivo tracers. In particular, the dual-labelled (111In-DTPA)n-trastuzumab-(IRDye800)m is capable of tracking HER2 overexpression in breast cancer patients [44]. Cetuximab has been repurposed for fluorescent imaging and it is in phase I and phase II clinical trials for malignant glioma and pancreatic cancer imaging and fluorescence-guided surgery with IRDye-800CW [45,46]. Furthermore, we previously provided preclinical evidence that the antibody arm which recognizes the hERG1 protein only (an scFv-hERG1) also selectively penetrates tumour masses [47]. While the scFvhERG1 can be applied to clinical settings but only after *local* injection (due to the concerns described in the Introduction), the scDb-hERG1-β1 can also be proposed for systemic in vivo use. Indeed, we showed that the scDb-hERG1-β1 lacks both specific (relating to heart) and systemic toxicity in mice, while maintaining good pharmacokinetic properties. The latter include effective biodistribution, good stability in serum and short half-life [31]. We also recently provided evidence that the scDb-hERG1-β1 does not affect the action potential duration (APD) in human cardiomyocytes (both atrial and ventricular) in vitro, and does not lengthen the QT interval, once intravenously injected into guinea pigs [48]. In particular, we showed that the scDb-hERG1-β1 could be used for molecular imaging, by conjugating it with ICG, i.e., a dye which is usually applied in PA imaging. PA imaging combines optical imaging with US technology. Chromophores such as ICG absorb light and emit sound waves which are detected by ultra-high frequency transducers. These signals are processed into high resolution photoacoustic images. The use of PA imaging in preclinical research has become increasingly widespread in recent years, particularly in oncology [49]. Recently, PA imaging has also been applied to numerous human diseases, usually for diagnostic purposes [50]. PA imaging uses endogenous contrast agents, such as oxygenated and deoxygenated haemoglobin, or exogenous contrast agents. The latter can be targeted for specific biomarkers, giving molecular and functional information on tumour microenvironment [49]. Finally, currently, the possibility of conjugating antibodies with PA contrast agents (e.g., ICG, Gold Nanoroads, Pan800) is being exploited for preclinical use [51,52].

The main result of the present paper is the good therapeutic efficacy of the combination between the scDb-hERG1-β1 with Gemcitabine. We showed a clear synergic effect of the antibody with the chemotherapeutic drug in vitro, which indicates not only that the combination of the two treatments can be favourable in PDAC, but that, as expected, the two compounds act on different pathways to trigger cell death in PDAC cells: a direct effect on DNA by Gemcitabine [29] and an effect on intracellular signalling by the antibody with a block in G1 (see Figure 3H in [31]). Furthermore, we showed that the Gemcitabine combination at sub-optimal doses has the same effects on tumour growth as Gemcitabine administered alone at a higher, therapeutic dose. What is more, the survival curves clearly showed that mice treated with either the scDb-hERG1-β1 alone or scDb-hERG1-β1 plus Gemcitabine survived longer than mice treated with Gemcitabine alone, clearly indicating a beneficial effect of the scDb-hERG1-β1 including its low toxicity. Indeed, although widely used for many years and often representing the first-line treatment for pancreatic cancer, several complications emerged with this chemotherapeutic drug. In fact, Gemcitabine has low therapeutic efficacy, due to its rapid clearance and its metabolic inactivation chargeable to cytidine deaminase. In order to obtain the desired cytotoxic effects, it is administered at high doses, which leads to severe side effects such as kidney failure, neutropenia and nausea [53].

## 5. Conclusions

Here, we provided evidence for a novel therapeutic strategy for PDAC, based on a single chain bispecific antibody (scDb-hERG1-β1, which targets a cancer-specific antigen, the hERG1/β1 integrin complex), in combination with low, sub-optimal doses of Gemcitabine. This combination showed good therapeutic efficacy with low toxicity resulting in improved survival time. Our results also raise the possibility of combining the scDb-hERG1-β1 with other chemotherapeutic drugs (e.g., FOLOFIRINOX) in neo-adjuvant therapy. Finally, given the excellent imaging data provided in this study, the scDb-hERG1-β1 can be exploited (e.g., in PA imaging) to identify patients that could benefit from the combination of the scDb-hERG1-β1 with chemotherapeutic drugs. Overall, our data pave the way for improving the therapy of PDAC, and possibly other cancers, by combining chemotherapy with ion channel modulators [54].

## 6. Patents

The hERG1/β1 integrin antibody has been patented by the University of Florence and licenced to MCK Therapeutics S.R.L. (under exclusive licence). MCK Therapeutics S.R.L. extended the patent internationally via PCT procedure, PCT/EP2018/06764 n., issued under no. WO2019/015936 on 24 January 2020 and granted in Australia, Canada, China, South Korea, United Arab Emirates, Eurasia, Japan, USA and Europe.

## Figures and Tables

**Figure 1 cancers-15-02013-f001:**
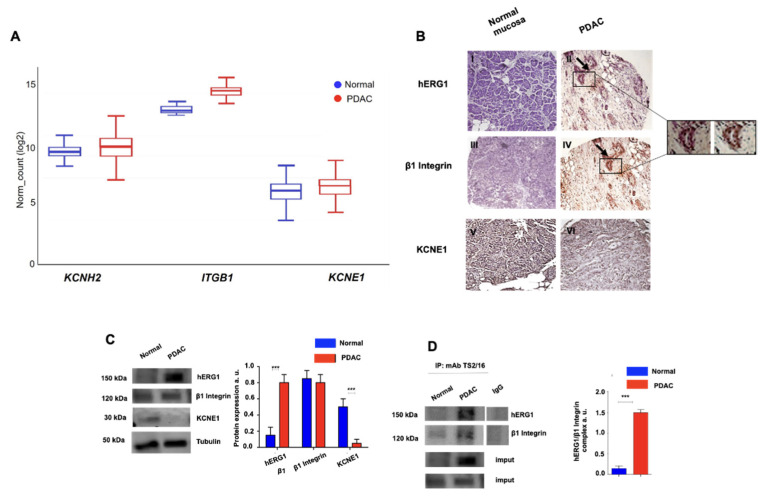
Gene expression profile and protein expression analysis of the hERG1/β1 integrin complex in pancreatic ductal adenocarcinoma samples. (**A**) Gene expression for KCNH2, ITGB1 and KCNE1 from TCGA/GTEx databases for PDAC and its normal counterpart. The final cohort featured 179 tumour samples from TCGA and 167 healthy samples from GTEx, for a total sample size of 346. The RNA-Seq expression data are given in units of log2 RSEM normalised counts. (**B**) Representative images of pancreatic samples. hERG1 in normal (I) and PDAC (II) samples (with representative insets), β1 integrin in normal (III) and PDAC (IV) samples (with representative insets), and KCNE1 in normal (V) and PDAC (VI) samples. Arrows indicate the staining of the same cells. IHC experiments and scoring assessment were performed as described in Materials and Methods. Scale bar: 200 μm. (**C**) hERG1, β1 integrin and KCNE1 protein expression (reported as normalized bands intensity; *n* = 3) in normal and PDAC samples. Representative blots of protein expression evolution (left panel) and densiometric analysis (right panel) are reported (*n* = 3). (**D**) Representative blots and corresponding densitometric analysis of the co-immunoprecipitation between hERG1 and β1 integrin (performed using TS2/16) in normal and PDAC samples. Data are presented as mean values ± s.e.m. *** *p* < 0.001.

**Figure 2 cancers-15-02013-f002:**
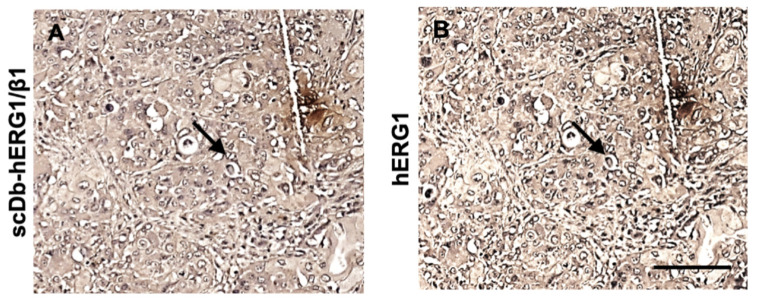
Immunohistochemistry analysis of hERG1 and hERG1-β1 complex expression in human PDAC samples. Representative images of a PDAC sample expressing the hERG1/β1 integrin complex (**A**) and hERG1 (**B**). Arrows indicate the staining of the same cells. Scale bar: 100 μm.

**Figure 3 cancers-15-02013-f003:**
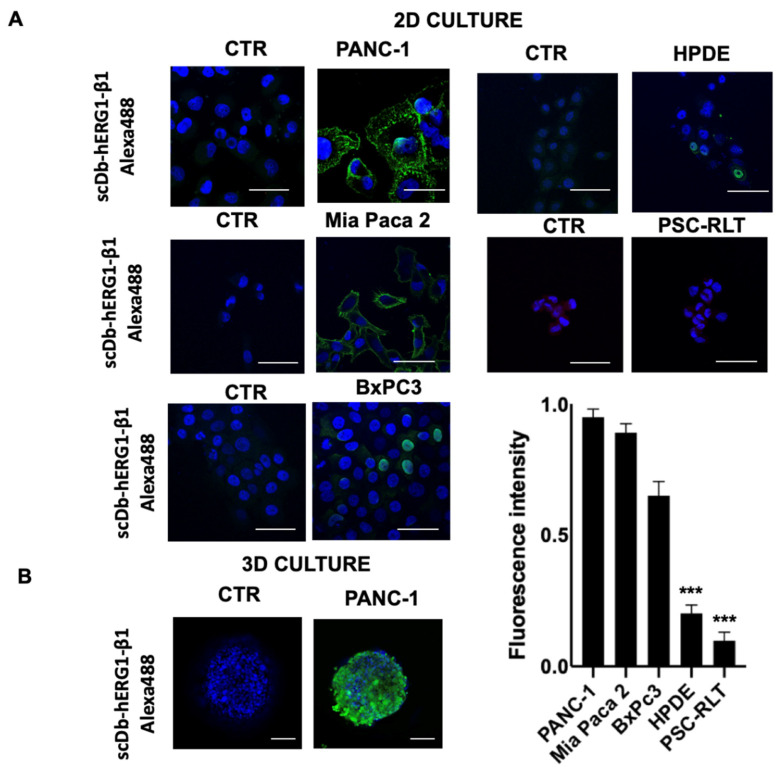
Expression of the hERG1-β1 integrin complex in PDAC and normal pancreatic cells determined by immunofluorescence (IF) and relative quantification. (**A**) Direct IF on fixed PANC-1, Mia Paca2, BxPC3, HPDE and PSC-RLT cells using the scDb-hERG1-β1-Alexa488 cultured in 2D. The picture labelled as “CTR” represents cells incubated with only the secondary antibody. Representative of three independent experiments performed in each cell line; the corresponding densitometric results are given in the bar graph on the right. *p* values were calculated using the Student’s *t* test. Scale bar = 100 μm. (**B**) Direct IF on fixed PANC-1 cells cultured as 3D using the scDb-hERG1-β1-Alexa488. The picture labelled as “CTR” represents cells incubated with only the secondary antibody. Representative of three independent experiments. Scale bar = 50 μm. *** *p* < 0.001.

**Figure 4 cancers-15-02013-f004:**
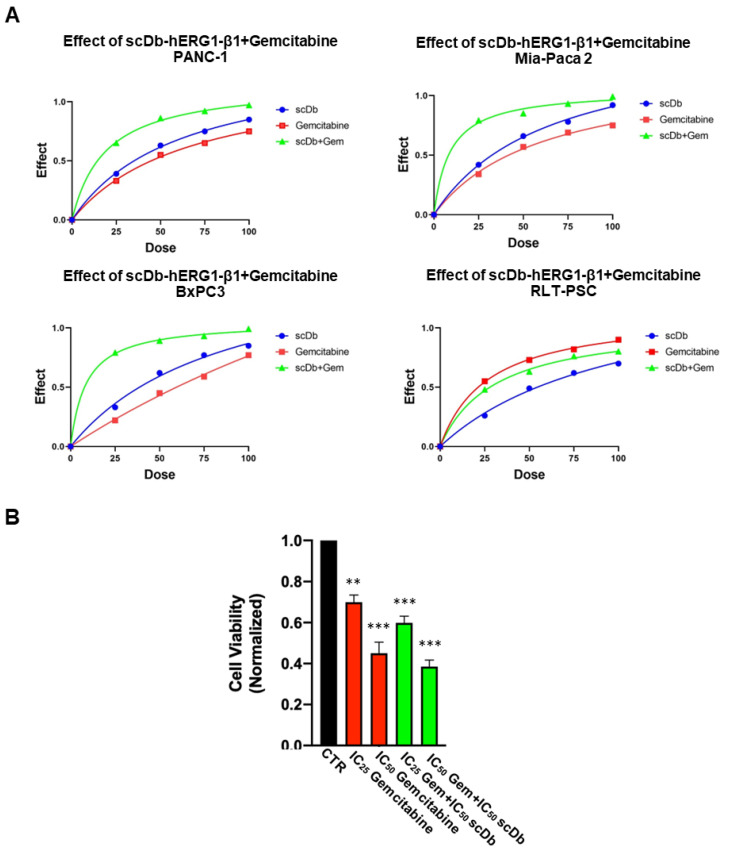
Effect of Gemcitabine and scDb-hERG1-β1 and of their combination on the vitality of different cell lines (PDAC and PSC-RLT). (**A**) Effects of Gemcitabine (red symbols and lines), scDb-hERG1-β1 (blue symbols and lines), and the combination of Gemcitabine and scDb-hERG1-β1 (green symbols and lines) on PANC-1, Mia Paca-2, BxPC3 and PSC-RLT cells. IC_25_, IC_50_, IC_75_ and IC_100_ doses of both Gemcitabine and scDb-hERG1-β1 were used. (**B**) Viability assay performed on PANC-1 cells using IC_25_ and IC_50_ Gemcitabine, IC_25_ Gemcitabine + IC_50_ scDb-hERG1-β1 and IC_50_ Gemcitabine+IC_50_ scDb-hERG1-β1. Ordinary one-way ANOVA with Dunnet’s multiple comparisons test was performed for statistical analysis ** *p* < 0.01, and *** *p* < 0.001.

**Figure 5 cancers-15-02013-f005:**
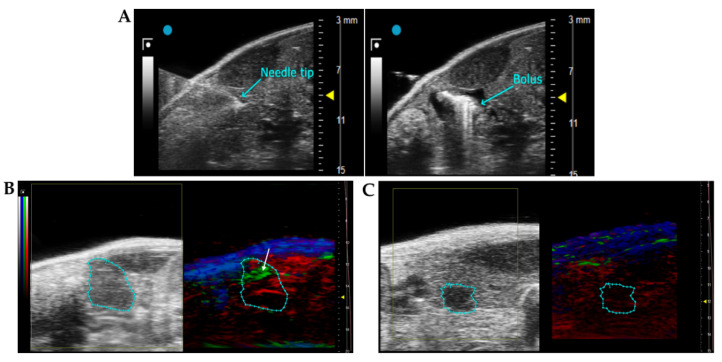
Injection procedure for obtaining the PDAC mouse model and imaging of the tumour masses in the pancreas. (**A**) Ultrasound acquisition during the US-guided injection on PANC-1 cells into the pancreas. In the left panel, the needle tip at the moment of injection is into the pancreatic tail. The correct injection of the bolus is checked by the presence of a bubble in the pancreas (right panel). (**B**,**C**) 2D US (left panels) and PA (right panels) images of tumour masses acquired 1 h after the intravenous injection of 16 mg/kg of scDb-hERG1-β1 conjugated with ICG (**B**) and 1 mg/kg ICG (**C**). The tumour is bordered by the cyan region of interest (ROI). In the right panels, obtained by PA imaging, red areas indicate the presence of oxyhaemoglobin, blue areas indicate the presence of deoxyhaemoglobin, and the green spots indicate the ICG signal (pointed out by the white arrow).

**Figure 6 cancers-15-02013-f006:**
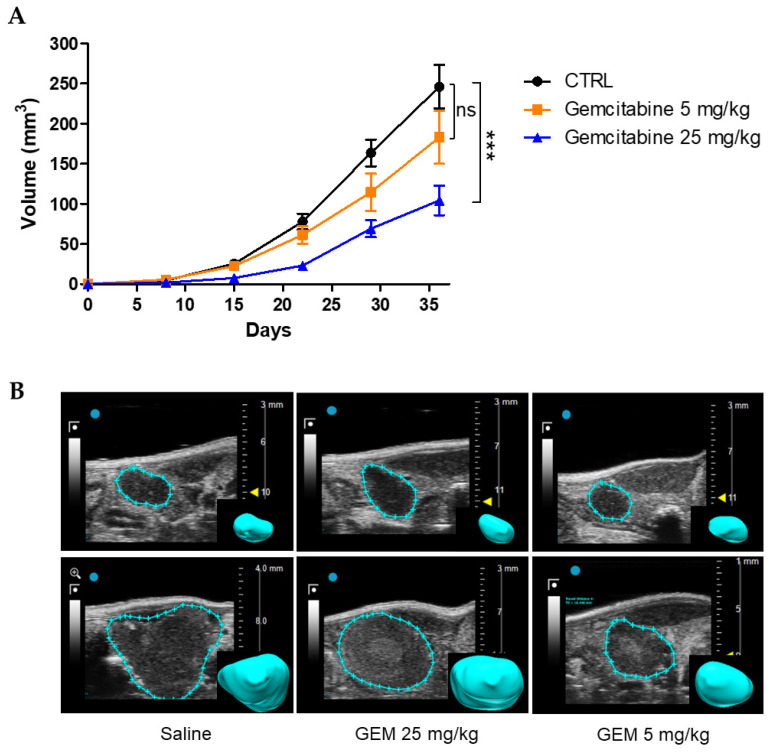
Effects of Gemcitabine in the orthotopic xenograft PDAC mouse model. (**A**) Time course of PDAC tumour growth in mice treated with saline (CTRL, *n* = 15) and different doses (25 mg/kg, *n* = 6 and 5 mg/kg, *n* = 8) of Gemcitabine. Tumour volume was measured with US imaging at several time points. Control group vs. Gemcitabine 25 mg/kg: *p* = 0.0004, and control group vs. Gemcitabine 5 mg/kg: *p* = 0.16, for Student’s *t* test on tumour volume at day 36. (**B**) Representative high-resolution US images of tumour masses at the beginning of the treatment (upper panels) and at the experimental end point (lower panels). The rendered 3D images of the tumour masses are shown in the insets. Data are presented as mean values ± s.e.m. *** *p* < 0.001; n.s.: not statistically significant.

**Figure 7 cancers-15-02013-f007:**
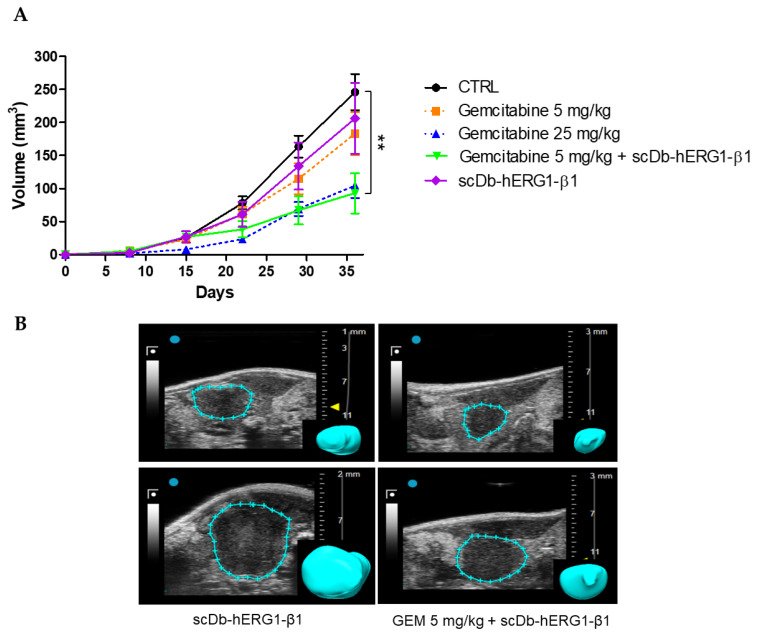
Effects of Gemcitabine in combination with the scDb-hERG1-β1 in the orthotopic xenograft PDAC mouse model. (**A**) Time course of PDAC tumour growth in mice treated with saline (CTRL), scDb-hERG1-β1 (16 mg/kg) and the combination Gemcitabine (5 mg/kg) and scDb-hERG1-β1 (16 mg/kg). The Gemcitabine 25 mg/kg and Gemcitabine 5 mg/kg groups are reported as dashed lines. Control (*n* = 15) vs. Gemcitabine 5 mg/kg scDb-hERG1-β1+ (*n* = 5) *p* = 0.003; control vs. ScDb-hERG1- β1 *p* = 0.5 (*n* = 9), Student’s *t* test. (**B**) Representative US images of tumour masses at the beginning of the treatment (upper panels) and at the end point (lower panels). The rendered 3D of the tumour masses are shown in the insets. Data are presented as mean values ± s.e.m. ** *p* < 0.01.

**Figure 8 cancers-15-02013-f008:**
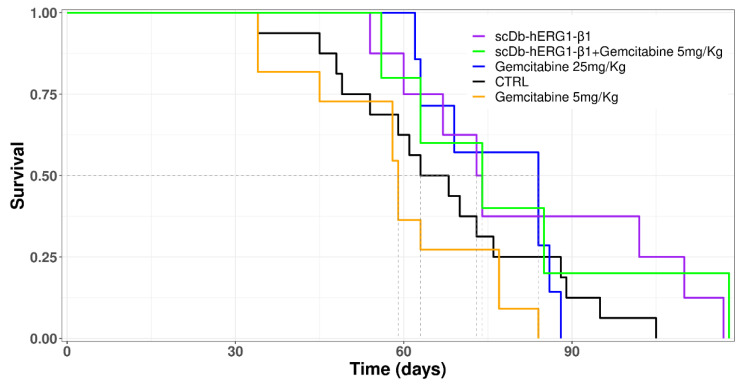
Kaplan–Meier survival curves for all groups with different treatments.

**Table 1 cancers-15-02013-t001:** hERG1, β1 integrin and KCNE1 expression in normal mucosa and pancreatic adenocarcinoma primary tissues. Absolute values and percentages (in parentheses) are indicated. R and *p* values of Pearson correlation coefficient are reported. Cut-off is defined as the number of positive cells. Statistically significant *p* values are in bold and underlined.

	Normal Mucosa				Ductal Adenocarcinoma		
	hERG1 −	hERG1 +	Pearson Correlation		hERG1 −	hERG1 +	Pearson Correlation
β1 Integrin −	23/40 (57.5%)	0/40 (0%)	R: −0.1594 *p*: 0.33	β1 Integrin −	5/40 (12.5%)	2/40 (5%)	R: 0.4835 ***p*: 0.001**
β1 Integrin +	17/40 (42.5%)	0/40 (0%)		β1 Integrin +	6/40 (15%)	27/40 (67.5%)	
	hERG1 −	hERG1 +	Pearson Correlation		hERG1 −	hERG1 +	Pearson Correlation
KCNE1 −	21/40 (52.5%)	0/40 (0%)	R: −0.0918 *p*: 0.58	KCNE1 −	9/40 (22.5%)	26/40 (65%)	R: −0.0954 *p*: 0.96
KCNE1 +	19/40 (47.5%)	0/40 (0%)		KCNE1 +	2/40 (5%)	3/40 (7.5%)	
	β1 Integrin −	β1 Integrin +	Pearson Correlation		β1 Integrin −	β1 Integrin +	Pearson Correlation
KCNE1 −	21/40 (52.5%)	0/40 (0%)	R: 1.0000 ***p* < 0.0001**	KCNE1 −	7/40 (17.5%)	28/40 (70%)	R: 0.2212 *p*: 0.19
KCNE1 +	2/40 (5%)	17/40 (42.5%)		KCNE1 +	0/40 (0%)	5/40 (12.5%)	

**Table 2 cancers-15-02013-t002:** IC_50_ values of Gemcitabine and scDb-hERG1-β1 on PANC-1, Mia Paca2, BxPC3 and PSC-RLT cells and respective combination index (CI) values. CI values were determined using Compusyn software.

Cell Line	IC_50_ Gemcitabine (μM)	IC_50_ scDb-hERG1-β1 (μg/mL)	Combination Index (CI)
PANC-1	40.66	18.13	0.97
MiaPaca-2	47.90	20.40	0.93
BxPC3	156.23	551.78	0.92
PSC-RLT	4.17	166.21	1.34

## Data Availability

Data are available upon request.

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
