# Peer review of "Combination Therapy with a Bispecific Antibody Targeting the hERG1/β1 Integrin Complex and Gemcitabine in Pancreatic Ductal Adenocarcinoma"

_cancers, 2023, doi:10.3390/cancers15072013_

Round 1

Reviewer 1 Report

Lottini et al. examined the combined effects of scDb-hERG1-b1 with gemcitabine on PDAC. Overall, the paper showed interesting data and scDb-hERG1-b1 supplementing low dose gemcitabine using an in vivo model and has a potential as a therapeutic strategy. However, the paper is not written concisely and often difficult to understand. Furthermore, the results section require more accurate interpretation of data (there are some over exaggeration of data). 

Please use more appropriate evaluations for the results section, many results are over exaggerated despite the effect is relatively low. I think that authors showed good data, there is no need to over exaggerate the data. 

Please recreate figure 1A. The font of description and axis information are too small and and resolution is too low.

Please remake Figure 5. Information is interesting but images and labels are too small and difficult to understand the data.

Please provide full images of western blots and provide information on the ladder sizes. The current images appear to be too cropped and no information on molecular sizes and lane information. It is very difficult to identify which lane was used for which figure. It would be ideal to provide uncut full blot images with complete information for transparency. 

Please perform statistical analysis for the Kaplan Meier curve. It would be good know which treatment group is statistically significant.

Author Response

Please use more appropriate evaluations for the results section, many results are over exaggerated despite the effect is relatively low. I think that authors showed good data, there is no need to over exaggerate the data. 

To comply with reviewer’s request, we have thoroughly corrected the Results section, de-emphasizing the results showed.

Please recreate figure 1A. The font of description and axis information are too small and and resolution is too low.

The panel A of the Figure 1 has been recreated, providing a better-quality resolution.

Please remake Figure 5. Information is interesting but images and labels are too small and difficult to understand the data.

We remade figure 5 enlarging the panels B and C in order to better show the signal obtained from the imaging. Moreover, we better described the panels B and C in the Figure legend, in order to better explain the photoacoustic images.

Please provide full images of western blots and provide information on the ladder sizes. The current images appear to be too cropped and no information on molecular sizes and lane information. It is very difficult to identify which lane was used for which figure. It would be ideal to provide uncut full blot images with complete information for transparency. 

To accomplish the reviewer’s request and to make the blots more understandable, new images of the membranes has been provided, with all the information about molecular size and lane identification.

Please perform statistical analysis for the Kaplan Meier curve. It would be good know which treatment group is statistically significant.

We performed statistical analysis on Kaplan Meier curve, and we observed statistically significance between the following groups: Gemcitabine 25mg/kg vs Gemcitabine 5 mg/kg and scDb-hERG1-β1 vs Gemcitabine 5 mg/kg. We added the statistical significance p value in main text (Line 562 and 571) and a new supplementary Table (Table S1) in supplementary section reporting pairwise comparison for Kaplan-Meier survival curve calculated with Long-Rank test was added

Reviewer 2 Report

The authors have done a interesting work on combination therapy for pancreatic ductal adenocarcinoma using a bi-specific antibody. However there are some minor comments.

1. The fig1A, Gene expression for KCNH2, ITGB1 and KCNE1 from  TCGA/GTEx databases for PDAC and its normal counterpart could presented well. The resolution of the current image is not readable.

2. The statistical analysis of fig4B, survivability data needs to be presented in the graph.

3. The use of ultrasound imaging to measure tumor volume is an excellent technique and authors have included them in their study which is impressive. They could also cite a literature in reference to it (https://doi.org/10.1021/acsami.1c21655) for using similar approach previously.

Author Response

  1. The fig1A, Gene expression for KCNH2, ITGB1 and KCNE1 from  TCGA/GTEx databases for PDAC and its normal counterpart could presented well. The resolution of the current image is not readable.

Figure 1 has been redrawn, providing a better quality resolution for panel A.

  1. The statistical analysis of fig4B, survivability data needs to be presented in the graph.

The statistical analysis has been added in the graph and also reported in the Figure Legend.

  1. The use of ultrasound imaging to measure tumor volume is an excellent technique and authors have included them in their study which is impressive. They could also cite a literature in reference to it (https://doi.org/10.1021/acsami.1c21655) for using similar approach previously.

We followed the reviewer’s suggestion adding a proper citation on the section Materials and Methods (line 285).

Round 2

Reviewer 1 Report

Thank you for addressing comments. No further comment.